# Better Drought Index between SPEI and SMDI and the Key Parameters in Denoting Drought Impacts on Spring Wheat Yields in Qinghai, China

**Miaolei Hou** [1,†], **Ning Yao** [1,†], **Yi Li** [1,*], **Fenggui Liu** [2,*], **Asim Biswas** [3], **Alim Pulatov** [4] **and Ishtiaq Hassan** [5]

1 Key Laboratory of Agricultural Soil and Water Engineering in Arid and Semiarid Areas, Ministry of Education, College of Water Resources and Architectural Engineering, Northwest Agriculture and Forestry University, Xianyang 712100, China; houmiaolei@nwafu.edu.cn (M.H.); yaoning@nwafu.edu.cn (N.Y.)
2 Academy of Plateau Science and Sustainability, Qinghai Normal University, Xining 810016, China
3 School of Environmental Sciences, University of Guelph, Guelph, ON N1G 2W1, Canada; biswas@uoguelph.ca
4 Tashkent Institute of Irrigation and Agricultural Mechanization Engineers, National Research University, Qoriy Niyoziy 39, Tashkent 100000, Uzbekistan; alimpulatov@mail.ru
5 Department of Civil Engineering, Capital University of Science and Technology (CUST), Islamabad 44000, Pakistan; eishtiaq88@cust.edu.pk
* Correspondence: liyi@nwafu.edu.cn (Y.L.); lfg_918@163.com (F.L.)
† These authors contributed equally to this work.

**Abstract:** Drought has great negative impacts on crop growth and production. In order to select appropriate drought indices to quantify drought influences on crops to minimize the risk of drought-related crops as much as possible, climate and spring wheat yield-related data from eight sites in the Qinghai Province of China were collected for selecting better drought index between standardized precipitation evapotranspiration index (SPEI, denoting meteorological drought) and soil moisture deficit index (SMDI, denoting agricultural drought) as well as the key parameters (timescale and month) in denoting drought impacts on spring wheat yields. The spring wheat yields during 1961–2018 were simulated by the DSSAT–CERES–Wheat model. Pearson correlations were used to investigate the relationship between SPEI and SMDI and between spring wheat yields and drought indices at different timescales. The results showed that: (1) SMDI reflected more consistent dry/wet conditions than SPEI when the timescales changed and (2) There were one- and two-month lags in SMDI compared to SPEI (with the higher correlation coefficients values of 0.35–0.68) during May to August and (3) May (the jointing period of spring wheat) and the two-month timescale of $SMDI_{0-10}$ (with the higher correlation coefficients values of 0.21–0.37) were key parameters denoting drought influences on spring wheat yield and (4) The correlations between the linear slopes of spring wheat yield reduction rate and linear slopes of $SMDI_{0-10}$ in May at the studied eight sites were considerable between 1961–2018 ($r = 0.85$). This study provides helpful references for mitigating the drought risk of spring wheat.

**Keywords:** soil moisture deficit index; standardized precipitation evapotranspiration index; Pearson correlation; DSSAT–CERES–wheat model; spring wheat





## 1. Introduction

Ongoing climate change has exacerbated the occurrence of different forms of drought dramatically [1,2], which is the largest climate-related threat to global agricultural production, especially in areas where crops depend solely on precipitation [3]. When a drought occurs without adequate irrigation, the shortage of water supply for crops occurs, which may cause crop growth hindrance, and then results in reduced crop yields, threatening food security [4,5]. Among different crops, wheat feeds about one-fifth of the population in the world [6], but its growth and maintaining stable production are facing more risks from

drought or other disasters [7]. Between 1980 and 2015, wheat yields reduced by 20.6% due to drought on a global scale [8]. It is of great significance for minimizing drought-related yield losses by studying the impact of drought on agriculture [9].

Drought represents an extended imbalance between water supply and demand [10]. Drought is commonly caused by insufficient precipitation and when soil moisture is insufficient to meet the needs of plant growth, which leads to agricultural drought following meteorological drought [11,12]. Many scholars have tried to characterize agricultural drought through disparate drought indices, including either meteorological or agricultural indices. There are some commonly used meteorological drought indices, such as Percentage of precipitation anomaly [13], Palmer drought severity index [14], Standardized precipitation index [15], Compound index, Relative humidity index, Reconnaissance drought index [16], Standardized precipitation evapotranspiration index (SPEI) [17], etc. Among them, SPEI not only has multi-timescale characteristics of a standardized precipitation index but also reflects the effects of global temperature change on drought, which is not only a good index for monitoring meteorological drought [18] but has also been widely used to monitor agricultural drought and analyze the impacts of crops due to drought [10,19,20]. For example, Pena-Gallardo et al. [21] found that there were significant correlations between wheat yields and the SPEI at timescales ranging from 1 to 18, particularly over the second half of the year in the counties of Eastern United States. Hamal et al. [1] discovered that the most correlated crop growth period for summer maize and winter wheat with SPEI was the sowing (February to May) and the growing period (November to February of the next year) across Nepal, respectively, which was the sensitive period of water deficit.

In agricultural drought monitoring, soil water plays a vital role [22]. Changes in soil water directly affect water availability, plant productivity, and crop yield [23,24]. The drought indices constructed based on soil moisture content have proved suitable for characterizing agricultural drought [25], which are Crop Moisture Index [26], Soil Moisture Percentage [27], Normalized Soil Moisture [28], and Soil Moisture Anomaly [29], etc. They have been widely used to identify and monitor agricultural droughts [30,31]. Higher-precision soil data products have enabled more agricultural drought monitoring indices to emerge [18,32,33]. Narasimhan and Srinivasan [34] developed a soil moisture deficit index (SMDI) based on the soil moisture simulated by SWAT (Soil and Water Assessment Tool) and found that wheat yield in the key growing period was highly correlated with SMDI. This index could reflect the short-term drought conditions in the root zone of the crop without seasonality, which has been used to monitor agricultural droughts in different regions after being proposed [35–37]. For example, Nepal et al. [38] found that the SMDI could reflect a better variation in drought conditions in the transboundary Koshi River basin (KRB) in the central Himalayan region and would be useful for the agricultural sector because it could provide a better understanding of soil moisture variation and agricultural droughts. Chen et al. [31] found that SMDI at a 0–10 cm depth had a greater impact on winter wheat yields from jointing to lactation. Wu and Li [39] pointed out that SMDI was more sensitive in recognizing early agricultural drought and performed better correlations between SMDI and winter and spring wheat yields variations across Russia during 1998–2013. These studies demonstrated that SMDI performed well in monitoring agricultural drought and could be used to indicate the effects of drought on crop yields.

Crop growth models can predict crop growth and yields by observing the physiological processes during crop growth with comprehensive genetic characteristics including crop varieties, climate, soil and management measures [40–42]. Some popularly used wheat growth models include the Agricultural and Food Research Council wheat model (AFRCWHEAT2) [43,44], General Large Area Model (GLAM–Wheat) [43], DSSAT–CERES–Wheat [45], Universal Crop Growth Simulator for water-limited conditions (SUCROS2) [44,46], etc. Among them, the DSSAT–CERES–Wheat model has been widely used to analyze the impact of the moisture deficit and climate variability on crop growth and yield [47–49], which has proved appropriate in simulating or predicting high-precision growth and yield of wheat in different regions or areas in the world [50,51]. The DSSAT–

CERES–Wheat model describes in detail the growth process of wheat, from seedling emergence to flowering, leaf emergence, grain filling, physiological maturity and harvest, based on the growth period of wheat by day step. This model can respond to many factors such as water stress, environment, crop genetic characteristics, pests and diseases, etc., and has been mainly used for agricultural yield forecast, production risk assessment and impact assessment of climate on agriculture [52].

China has suffered from drought hazards for a long history [18,53]. The average agricultural drought-affected areas in China exceeded 24.43 million hm$^2$, and the annual food loss caused by drought reached 30 billion kg, accounting for more than 60% of the total loss from natural disasters (www.mwr.gov.cn/, accessed on 8 December 2021). Wheat is planted mostly in north China (colder than the south) and accounts for more than 20% of China's grain crops. However, most of the wheat-planting regions in China including Qinghai have been threatened by drought disasters [54]. Spring wheat is an important food crop with a planting area of $11.16 \times 10^4$ hm$^2$ (tjj.qinghai.gov.cn/, accessed on 20 June 2020). The farmland in Qinghai is mostly rain-fed, so the natural precipitation directly affects agricultural production to a great extent. There is a high risk of drought during spring wheat growth. If drought occurs in spring or summer, it would be unfavorable to wheat emergence and growth and affect the later stage of wheat grain production [55]. Therefore, it is significant to investigate drought impacts on spring wheat growth and yields in Qinghai, China in order to reduce yield losses.

Previous studies combined different drought indices with crop growth or yield-related indices to study the impact of drought on crop yields, but only a single meteorological or agricultural drought index was applied. Few studies have compared different drought indices and their appropriateness for monitoring drought conditions and drought impacts on agriculture and crops. The objectives of this study were (1) to compare the differences and connections between the meteorological drought index (SPEI) and the agricultural drought index (SMDI) at 0–10 cm and 10–40 cm depths in monitoring the drought at each growth stage of spring wheat in Qinghai province, China; (2) to identify the more appropriate drought index, the key month on spring wheat growth, and the key timescale of the selected better drought index that denote a closer relationship between spring wheat growth yields and drought severity based on correlations between the spring wheat growth and drought indices; and (3) to analyze the relationship between yield reduction rate calculations based on selected drought index and wet and dry conditions at the key month under the key timescale.

## 2. Data and Methodology

### 2.1. Study Area and Selected Sites

Qinghai province is located in the northwestern region of China, with an area of approximately 720,000 km$^2$, accounting for 7.5% of the total Chinese area. Qinghai is an arid and semi-arid area with an annual average temperature of 2–9 °C, a frost-free period of 100–200 days, and annual average precipitation of 25–500 mm [56]. There is large variability in precipitation, with higher precipitation in the southeast and lower precipitation in the northwest. Spring wheat is one of the main food crops in Qinghai Province.

The 8 selected sites with spring wheat planting are located in the Qaidam Basin and the eastern agricultural region of Qinghai province, China. These sites have suffered from drought and flood hazards in the past which affected local agricultural production. The distribution of the selected sites and elevation is mapped in Figure 1.

### 2.2. Collection of Climate, Soil, and Crop Data

The observed climate data were obtained from China meteorological administration data network (data.cma.cn, accessed on 30 December 2019), including daily precipitation ($P$), relative humidity ($RH$), daily maximum temperature ($T_{max}$), daily minimum temperature ($T_{min}$), wind speed at 2 m height ($U$) and sunshine hours from 1961–2018 at the 8 studied sites.

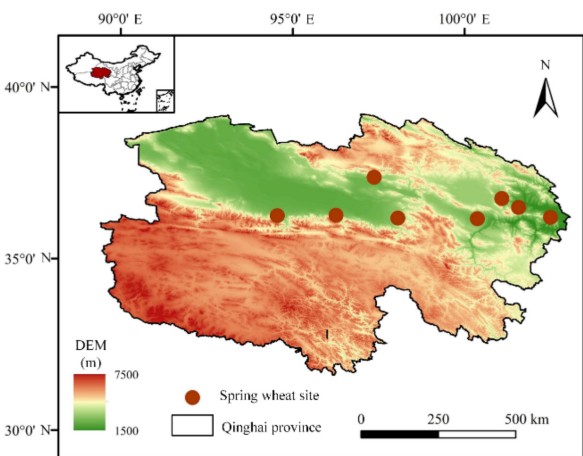

**Figure 1.** Spatial distribution of elevation and spring wheat sites in Qinghai of China.

The soil property data including saturated moisture content (SAT), soil water content at the permanent wilting point (WP), soil water content at the field capacity (FC) and saturated hydraulic conductivity (SHC) were obtained from the Chinese soil moisture dataset with a spatial resolution of 30 × 30 arc seconds [57]. The field capacity data have 7 depth ranges of 0–4.5, 4.5–9.1, 9.1–16.6, 16.6–28.9, 28.9–49.3, 49.3–82.9 and 82.9–138.3 cm, respectively. The average layer soil moisture (at 4 layers of 0–10 cm, 10–40 cm, 40–100 cm and 100–200 cm) from 1961–2018 were collected from Global land data assimilation system–Noah–simulated (GLDAS–Noah) with a spatial resolution of 0.25° × 0.25°, which has good applicability in Qinghai through comparative analysis with observed data. The GLDAS soil moisture was converted to volumetric values ($m^3 \ m^{-3}$).

The spring wheat growth period and yield data (2001–2013) for the 8 sites were collected from Qinghai institute of meteorological science (qh.cma.gov.cn, accessed on 30 December 2020) and agricultural meteorological data on China meteorological administration data network (data.cma.cn, accessed on 30 December 2019). The agricultural disaster data were collected from National Qinghai–Tibet Plateau Data Center (data.tpdc.ac.cn, accessed on 5 January 2021). According to the observed data, the whole growth period of spring wheat is divided into 5 physiological stages, namely sowing–emergence stage, emergence–jointing stage, jointing–earing stage, heading–milk maturity, and milk maturity–maturity stage. The start and end dates of spring wheat growth period at the 8 sites are presented in Table 1. The multi-year average of the growth period days is used to represent the local general growth period. The full growth period of spring wheat is from March to August.

**Table 1.** Spring wheat growth period at 8 sites of Qinghai, China.

| Site | Growth Period | | | | | |
|------|------|------|------|------|------|------|
| | Sowing | Emergence | Jointing | Heading | Milk Maturity | Maturity |
| Minhe | 28 Mar. | 09 Apr. | 17 May | 12 Jun. | 09 Jul. | 26 Jul. |
| Ledu | 11 Mar. | 05 Apr. | 19 May | 10 Jun. | 15 Jul. | 30 Jul. |
| Huangyuan | 31 Mar. | 26 Apr. | 07 May | 02 Jul. | 07 Jul. | 03 Sep. |
| Huzhu | 01 Apr. | 23 Apr. | 10 Jun. | 05 Jul. | 04 Aug. | 02 Sep. |
| Guide | 07 Mar. | 02 Apr. | 17 May | 10 Jun. | 15 Jul. | 30 Jul. |
| Datong | 28 Mar. | 22 Apr. | 05 Jun. | 01 Jul. | 05 Aug. | 01 Sep. |
| Xunhua | 28 Feb. | 28 Mar. | 14 May | 30 May | 30 Jun. | 22 Jul. |
| Tongren | 20 Mar. | 13 Apr. | 03 May | 24 Jun. | 24 Jul. | 15 Aug. |

*2.3. Computation of Drought Indices*

2.3.1. Standardized Precipitation Evapotranspiration Index (SPEI)

The SPEI is calculated by the difference in precipitation (*P*) and reference crop evapotranspiration ($ET_0$) for each site. The Penman–Monteith formula is recommended by the

Food and Agriculture Organization of the United Nations (FAO) and has proved to have good performance in different parts of the world [58]. The procedure for SPEI computation follows [17]:

To calculate the $ET_0$ at the monthly timescale:

$$ET_0 = \frac{0.408\Delta\, R_n - G\, + \Delta \frac{900}{T+273} U_2(e_n - e_a)}{\Delta + \gamma(1 + 0.34U_2)} \tag{1}$$

where $R_n$ is the net radiation (MJ m$^{-2}$ d$^{-1}$); $T$ is the daily average temperature (°C); $U_2$ is the wind speed at 2 m (m s$^{-1}$); $e_s$ and $e_a$ are the saturated and actual water vapor pressure (kPa); $\Delta$ is the slope of the saturated water vapor pressure-temperature curve (kPa/°C); $\gamma$ is the wet and dry meter constant (kPa/°C).

To calculate the difference between $P$ and $ET_0$, namely $D$:

$$D = P - ET_0 \tag{2}$$

To normalize the data series $D$. Vicente-Serrano, et al. [17] compared the fitting effects of Log-logistic, Pearson III, Lognormal, and generalized extreme values on the sequence. The results showed that Log-logistic distribution showed better performance:

$$F(x) = \left[ 1 + \left( \frac{\alpha}{x - \gamma} \right)^{\beta} \right]^{-1} \tag{3}$$

where $F(x)$ is the cumulative probability distribution function for a given time scale; $\alpha$ is the scale parameter; $\beta$ is the shape parameter; $\gamma$ is the origin parameter, which can be obtained by fitting the linear moment. The parameters are calculated as follows:

$$\alpha = \frac{(\omega_0 - 2\omega_1)\beta}{\Gamma(1 + 1/\beta)\Gamma(1 - 1/\beta)} \tag{4}$$

$$\beta = \frac{2\omega_1 - \omega_0}{6\omega_1 - \omega_0 - 6\omega_2} \tag{5}$$

$$\gamma = \omega_0 - \alpha\Gamma(1 + 1/\beta)\Gamma(1 - 1/\beta) \tag{6}$$

where $\Gamma$ is the factorial function; $\omega_0$, $\omega_1$, $\omega_2$ is the probability-weighted distance of the original data sequence $D$, the calculation method is:

$$\omega_i = \frac{1}{N}\sum_{i=0}^{N}(1 - F_i)^s D_i \tag{7}$$

$$F_i = \frac{i - 0.35}{N} \tag{8}$$

where $N$ is the number of months involved in the calculation.

To standardize the cumulative probability density:

$$P(D) = 1 - F(x) \tag{9}$$

When $P(D) \leq 0.5$:

$$W = \sqrt{-2\ln P(D)} \tag{10}$$

$$\text{SPEI} = W - \frac{c_0 + c_1 W + c_2 W^2}{1 + d_1 W + d_2 W^2 + d_3 W^3} \tag{11}$$

When $P(D) > 0.5$, $P(D)$ is replaced by $1 - P(D)$. Here $c_0 = 2.515517$, $c_1 = 0.802853$, $c_2 = 0.010328$, $d_1 = 1.432788$, $d_2 = 0.189269$ and $d_3 = 0.001308$.

### 2.3.2. Soil Moisture Deficit Index (SMDI)

SMDI is calculated at two soil depths of 0–10 cm and 10–40 cm by using soil moisture content data at 1- to 6-timescales from 1961 to 2018 [34]. The calculation procedure is as follows: (1) Using the long-term median, maximum and minimum soil moisture at a certain timescale to calculate the percentage soil moisture deficit, as follows:

$$SD_{i,j} = \begin{cases} \frac{SW_{i,j} - MSW_j}{MSW_j - \min SW_j} \times 100, SW_{i,j} \le MSW_j \\ \frac{SW_{i,j} - MSW_j}{\max SW_j - MSW_j} \times 100, SW_{i,j} > MSW_j \end{cases} \tag{12}$$

where $SD_{i,j}$ is the soil moisture deficit in the $j$th month of the $i$th year (%); $SW_{i,j}$ is the mean soil moisture content at a certain timescale in the soil profile (mm); $MSW_j$ is the long-term median soil moisture content in the soil profile (mm); max$SW_j$ is the long-term maximum soil moisture content in the soil profile (mm); min$SW_j$ is the long-term minimum soil moisture content in the soil profile (mm) ($i$ =1, 2, . . . , 58, and $j$ = 1, 2, . . . , 12).

(2) By using formula (13), the seasonality inherent in soil moisture is removed. Hence, SD is compared across seasons. To determine drought severity, the main challenge is to choose the time step over which the dryness values are accumulated [22]. Thus, the drought index is calculated on an incremental basis as suggested by Palmer [14]:

$$\text{SMDI}_{i,j} = \begin{cases} \frac{SD_{i,j}}{50}, j = 1 \\ 0.5\text{SMDI}_{i,j-1} + \frac{SD_{i,j}}{50}, j > 1 \end{cases} \tag{13}$$

In order to compare SMDI and SPEI, the step is modified to 2 during calculation, and the range of SMDI is re-adjusted from (−4 to 4) to (−2 to 2) to be consistent with the SPEI value. The value of SPEI or SMDI in a certain month (or a certain timescale) is the average value of the previous months till the current month. For example, the SPEI or SMDI in May at the 2-month timescale is an average value of SPEI or SMDI from April to May. The dry or wet conditions classification using SPEI and SMDI are shown in Table 2.

**Table 2.** Dry/wet condition classification based on SPEI and SMDI.

| Dry/Wet Severity Level | Range of SPEI | Range of SMDI |
|---|---|---|
| Extreme wet | $2 \le$ SPEI | $2 \le$ SMDI |
| Severe wet | $1.5 \le$ SPEI $< 2$ | $1.5 \le$ SMDI $< 2$ |
| Moderate wet | $1 \le$ SPEI $< 1.5$ | $1 \le$ SMDI $< 1.5$ |
| Mild wet | $0.5 \le$ SPEI $< 1$ | $0.5 \le$ SMDI $< 1$ |
| Normal | $-0.5 <$ SPEI $< 0.5$ | $-0.5 <$ SMDI $< 0.5$ |
| Mild dry | $-1 <$ SPEI $\le -0.5$ | $-1 <$ SMDI $\le -0.5$ |
| Moderate dry | $-1.5 <$ SPEI $\le -1$ | $-1.5 <$ SMDI $\le -1$ |
| Severe dry | $-2 <$ SPEI $\le -1.5$ | $-2 <$ SMDI $\le -1.5$ |
| Extreme dry | SPEI $\le -2$ | SMDI $\le -2$ |

### 2.4. Crop Growth and Yield Simulation Using the DSSAT–CERES–Wheat Model

Since the observed crop yield data at the 8 studied sites are only for 2 to 13 years (2001–2013), in order to study the long-term effects of drought on spring wheat yield, The DSSAT–CERES–Wheat model is used to extend the growth and yield sequence to 1961–2018, which is a sub-module in DSSAT–CERES–wheat used for simulating the growth process of spring wheat. The input data include four modules: meteorology, soil parameters, yield management, and crop genetic coefficient. The input meteorological data mainly include solar radiation, $T_{\max}$, $T_{\min}$, $P$, $U$ and sunshine hours. The input soil parameters include SAT, WP, FC, SHC, and field management data include sowing period, fertilization amount, irrigation method and irrigation amount, etc.

In DSSAT–CERES–wheat, the generalized likelihood uncertainty estimation (GLUE) is used to debug the genetic coefficients of spring wheat, which include P1V, P1D, P5, G1,

G2, G3 and PHINT. The debugging process of crop genetic parameters is divided into two rounds, each 6000 times. In the first round, the crop phenological parameters are adjusted and in the second round, the growth parameters of crops are estimated.

The debugging results of spring wheat genetic parameters at the 8 selected sites are shown in Table 3. Then, the anthesis date, maturity date and yield data of the first 6 years (2001–2006) are used to correct the genetic coefficients, and the genetic coefficients are validated with the data of the next 7 years (2006–2013). After assessing the calibration and validation performance of DSSAT–CERES–wheat, the corrected genetic coefficients are used to simulate the crop's leaf area index (LAI) and meteorological yields of spring wheat from 1961 to 2018.

**Table 3.** Genetic coefficients of spring wheat at the selected 8 sites.

| Site | Spring Wheat Parameter | | | | | | |
|---|---|---|---|---|---|---|---|
| | P1V | P1D | P5 | G1 | G2 | G3 | PHINT |
| Deling | 19.72 | 38.59 | 783.8 | 16.76 | 37.68 | 1.764 | 72.56 |
| Geer | 19.70 | 33.76 | 799.9 | 17.06 | 45.98 | 1.731 | 61.30 |
| Dulan | 19.12 | 39.65 | 789.8 | 19.88 | 23.22 | 1.570 | 95.08 |
| Gonghe | 19.57 | 37.65 | 796.4 | 17.87 | 21.58 | 1.205 | 63.25 |
| Minhe | 19.91 | 38.34 | 798.8 | 24.15 | 26.96 | 1.616 | 69.65 |
| Nuomu | 19.90 | 34.61 | 773.0 | 19.04 | 23.20 | 1.820 | 63.35 |
| Huangyuan | 19.67 | 38.67 | 786.1 | 19.74 | 20.89 | 1.942 | 61.40 |
| Huzhu | 19.79 | 38.66 | 792.8 | 15.97 | 20.46 | 1.488 | 67.25 |
| Mean | 19.67 | 37.49 | 790.08 | 18.81 | 27.50 | 1.64 | 69.23 |
| Standard deviation | 0.25 | 2.12 | 9.00 | 2.59 | 9.34 | 0.23 | 11.20 |
| Coefficient of variation/% | 1.28 | 5.67 | 1.14 | 13.75 | 33.97 | 13.91 | 16.18 |

The coefficient of determination ($R^2$) and relative root mean square error (*RRMSE*) are used to evaluate the performance of DSSAT–CERES–wheat during calibration, validation and simulation processes. The equations of $R^2$ and RRMSE are as follows:

$$R^2 = \left( \frac{\sum\limits_{m=1}^{n} (o_m - \bar{o})(s_m - \bar{s})}{\sqrt{\sum\limits_{m=1}^{n} (o_m - \bar{o})^2} \sqrt{\sum\limits_{m=1}^{n} (s_m - \bar{s})^2}} \right) \tag{14}$$

$$RRMSE = \frac{\sqrt{\frac{1}{n} \sum\limits_{m=1}^{n} (s_m - o_m)^2}}{\bar{o}} \tag{15}$$

where $o_m$ is the observed value in the $m$th year ($m$ = 1, 2, . . . , 13), $\bar{o}$ is the mean value of $o_m$, $s_m$ is the simulated value in the $m$th year, $\bar{s}$ is the mean value of $s_m$, and $n$ is the number of samples. Generally, the higher the $R^2$ values, or the lower the RRMSE values, the better performance of DSSAT–CERES–wheat.

### 2.5. Correlations between Yields and Drought Indices

The Pearson correlation coefficient ($r$) is used to evaluate the relationships between (i) SPEI and SMDI in current month or lagged for 1 to 2 months; and (ii) SPEI (or SMDI) and yields (or $LAI_{max}$) at the 1- to 6- month timescales during the entire growth period of spring wheat. The range of $r$ is between −1 and 1. The larger the absolute value of $r$, the closer the relationship between two variables.

### 2.6. Yield Reduction Rate of Spring Wheat

When SPEI (or SMDI) is between −0.5 and 0.5, it is regarded as a normal year without drought or waterlogging, and the average yield of the normal year is considered the reference yield. The reference yield is calculated as follows:

$$Yield_{k,normal} = \frac{Yield_{k,1} + Yield_{k,2} + \cdots Yield_{k,n}}{n} \tag{16}$$

where $Yield_{k,normal}$ represents the reference yield of spring wheat at the $k$th site ($k$ = 1, 2, ... , 8), and $n$ is the number of normal years at the site.

The annual reduction rate is calculated as follows:

$$YRR_{k,i} = \frac{Yield_{k,i} - Yield_{k,normal}}{Yield_{k,normal}} \times 100\% \tag{17}$$

where $YRR_{k,i}$ represents the yield reduction rate of spring wheat at the $k$th site, and $i$ is the number of years ($i$ = 1, 2, ... , 58). $Yield_{k,i}$ represents the actual yield in the $i$th year. When $YRR_{k,i}$ is negative, it means that production has been reduced in the $i$th year.

Python (version 3.3.9) is applied to analyze data and draw figures.

The flow chart which illustrates the main methodology and idea of this research is shown in Figure 2.

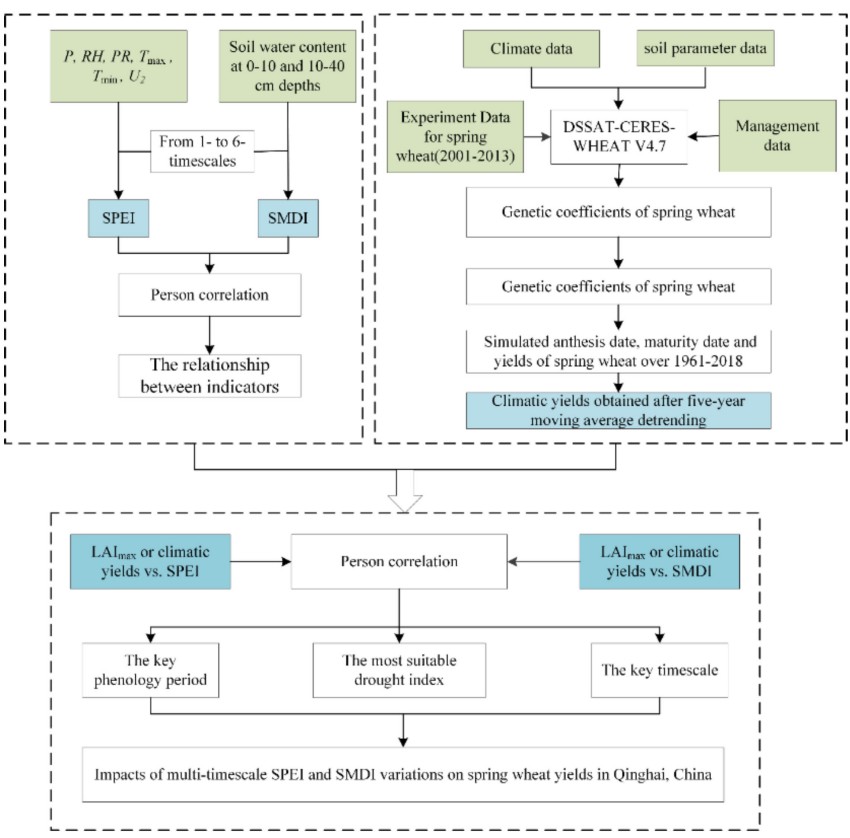

**Figure 2.** The overall technology roadmap of the research.

## 3. Results

### 3.1. The Drought Variations Indicated by SPEI and SMDI

A total of eight sites were selected for this research. However, there was a great amount of data and the characteristics of the studied variables were similar; therefore, here the Gonghe site is taken as the example for it is representative of the selected 8 sites. The temporal variations of monthly $P$, $ET_0$, $P - ET_0$ and soil moisture deficit in 1961–2018 are compared in Figure 3. The results showed that $P$ and $ET_0$ have relatively similar patterns

of variation within years, $P$ (or $ET_0$) increased from April or May and reached peaks in July or August. Monthly $P - ET_0$ ranged between −75 mm to 50 mm and were largely negative, indicating Gonghe suffers long-term drought. Soil moisture deficit varied more randomly than $P$ and $ET_0$, but the values were generally higher from May to August than the other months. The other 7 sites showed similar annual patterns, but there were some regional differences in precipitation (Figure S1), with sites located in the west of Qinghai province (Deling, Geer, Nuomu, Geer) having less rainfall than the eastern sites (Huangyuan, Gonghe, Huzhu, Minhe).

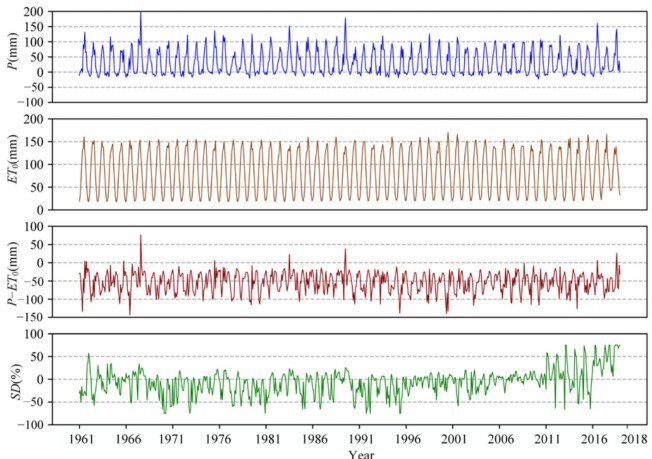

**Figure 3.** Variations of monthly $P$, $ET_0$, $P - ET_0$ and soil moisture over 1961–2018 at Gonghe site.

The temporal variations of SPEI, $SMDI_{0-10}$, and $SMDI_{10-40}$ at the 1- to 6-month timescales during the spring wheat growing season of 1961 to 2018 at Gonghe site are illustrated in Figure 4 (The other 7 sites are shown in Figure S2).

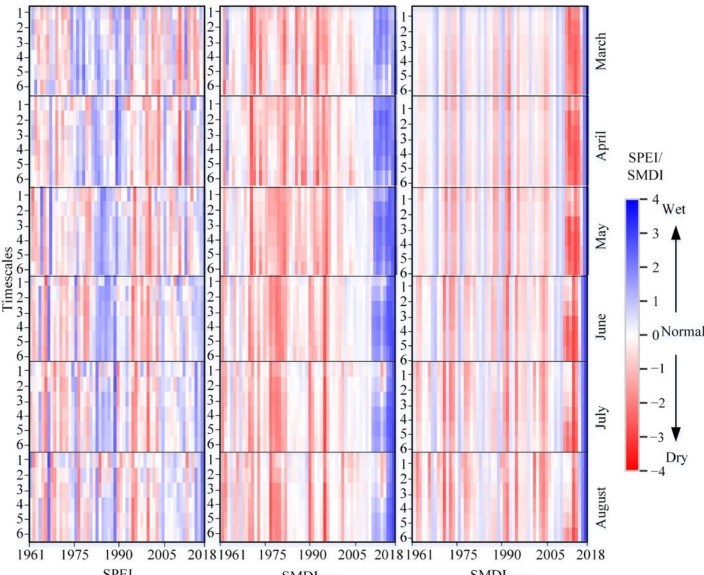

**Figure 4.** Temporal variations of the monthly SPEI, $SMDI_{0-10}$ and $SMDI_{10-40}$ at the 1- to 6-month timescales during the growing seasons of spring wheat at the Gonghe site.

The results in Figure 4 showed that (1) SPEI varied more randomly than SMDI over 1961–2018, with a wetter period over 1980–1989. From $SMDI_{0-10}$, there was a long drier period from 1970–1996 but a short wetter period from 2010–2018. Variations of $SMDI_{10-40}$ were generally similar to $SMDI_{0-10}$ but looked wetter over the entire studied period, with a typically different short drier period of 2011–2016, and (2) at the 1- to 6-month

timescales, SMDI (0–10 and 10–40 cm) during the growth stage of spring wheat varied similarly, reflecting generally consistent dry or wet conditions. Variations in SPEI at different timescales were less similar than those in SMDI and (3) in different years and months, the drought levels identified by SPEI and SMDI (0–10 cm and 10–40 cm) were not consistent. For example, the year 2001 was identified as a severe drought by SPEI, but a normal year according to SMDI (both at 0–10 cm and 10–40 cm depth), while 1991 was identified as a normal year by SPEI but a moderate drought year by SMDI (0–10 cm and 10–40 cm).

In fact, different types of drought originate from meteorological drought but, due to the different processes of drought formation, there is a phase difference in time. In order to investigate the relationship between meteorological drought and agricultural drought, the $r$ values between SPEI and $SMDI_{0-10}$ with a lagged time of 0–5 months during the spring wheat growth period were calculated on the 1- to 6- month timescales (Similar to $SMDI_{0-10}$ vs. $SMDI_{10-40}$ with a lagged time of 0–5 months and SPEI vs. $SMDI_{0-10}$ with a lagged time of 0–5 months). The result showed that SPEI was more closely related to $SMDI_{0-10}$ than $SMDI_{10-40}$ on each timescale, indicating that the surface soil moisture was easily affected by precipitation, temperature and other factors (Figure 5). From March to April (the sowing–emergence period of spring wheat), there is no obvious correlation ($-0.23$–$0.27$) between SPEI and $SMDI_{0-10}$ $SMDI_{10-40}$ in the current month or time lag of 1–5 months from March to April, suggesting that the source of soil moisture is not dependent on atmospheric precipitation at this time and that alpine snowmelt may also have a partial influence. While during May to June (the jointing—heading period of spring wheat), SPEI generally had a higher $r$ value with 1 and 2 months lagged $SMDI_{0-10}$ (0.3–0.59) and $SMDI_{10-40}$ (0.31–0.45). In July (the milking maturity period of spring wheat), SPEI showed a higher correlation with a 1-month lag between $SMDI_{0-10}$ and $SMDI_{10-40}$. The results indicated that there is a certain hysteresis agricultural drought (indicated by SMDI) compared with meteorological drought (indicated by SPEI) when precipitation increases, and the correlation between surface soil moisture and SPEI was higher than deep.

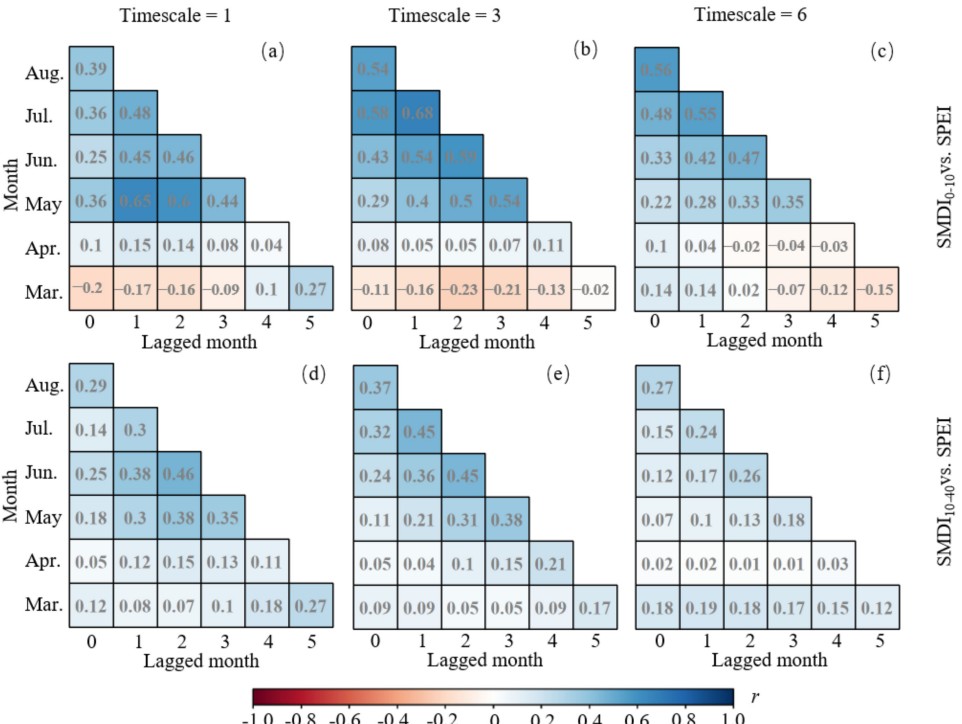

**Figure 5.** The $r$ values between SPEI and $SMDI_{0-10}$ (**a–c**) or SPEI and $SMDI_{10-40}$ (**d–f**) at the 1, 3 and 6-month timescales with lagged time of 0 to 5 months during growth period of spring wheat (March to August).

The coincidence between drought events indicated by SPEI and SMDI compared with the actual drought events is identified by Huangyuan, Huzhu and Minhe, where historical data are relatively complete (Figure 6). The results showed that $SMDI_{0-10}$ among the three indices had the highest agreement with actual drought conditions (43–82%), followed by $SMDI_{10-40}$ (39–60%) and SPEI (29–43%), respectively. This may be due to the fact that most of the actual drought records were related to rainfall deficits, damage to farmland, and reduced grain production.

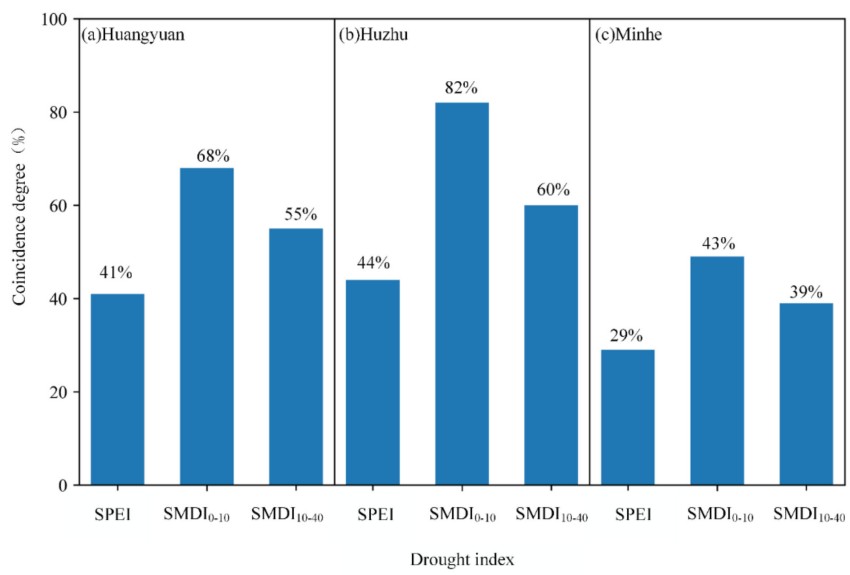

**Figure 6.** Coincidence degrees of identified drought events based on the selected drought indices.

### 3.2. Spring Wheat LAI and Yield Simulated by DSSAT-CERES-Wheat Model

3.2.1. Performance of DSSAT–CERES–Wheat Model

The performance of the DSSAT–CERES–wheat model has been evaluated by plotting the simulated and observed anthesis date, maturity date, and yield values of spring wheat for both calibration and validation processes (Figure 7), and the $R^2$ and RRMSE were used to verify whether the simulated value is consistent with the observed value. The results showed that the higher $R^2$ ($0.70 < R^2 < 0.85$) and lower RRMSE ($0.04 < RRMSE < 0.18$) between simulated and observed values during the calibration and verification process indicated a better accuracy of the model simulation. Therefore, the calibrated genetic parameters of spring wheat could be further used for the simulation of crop growth and yield over the period 1961–2018 at the eight studied sites.

3.2.2. Annual Variations of Spring Yields

The annual variations of simulated (climatic) yields in 1961–2018 for the eight sites are shown in Figure 8. There were generally random fluctuations of actual yields at the eight sites over 1961 to 2018, and there was a relatively large fluctuation between 2001 and 2011. The yields of spring wheat ranged from 2050 to 12540 kg ha$^{-1}$, and higher yields from 2005 to 2010 were shown. There was a site rank of mean trend yield: Geer > Deling > Minhe > Nuomu > Huangyuan > Gonghe > Dulan > Huzhu.

### 3.3. The Effects of Drought on Spring Wheat Growth and Yields

Values of $r$ for drought indices (SPEI or $SMDI_{0-10}$) vs. spring wheat growth indices ($LAI_{max}$ or climatic yield) are illustrated for the site Gonghe in Table 4 (The $r$ values between drought indices and spring wheat growth indices of the other 7 sites are shown in Table S1 and Table S2 respectively). The correlation between SPEI and spring wheat climatic yield was generally low ($-0.11 < r < 0.17$), while there was mostly a negative correlation with $LAI_{max}$ ($-0.26 < r < 0.11$). However, $SMDI_{0-10}$ was positively correlated

with $LAI_{max}$ and climatic yields ($-0.01 < r < 0.37$), indicating different and complicated effects of meteorological and agricultural droughts on spring wheat growth and yields.

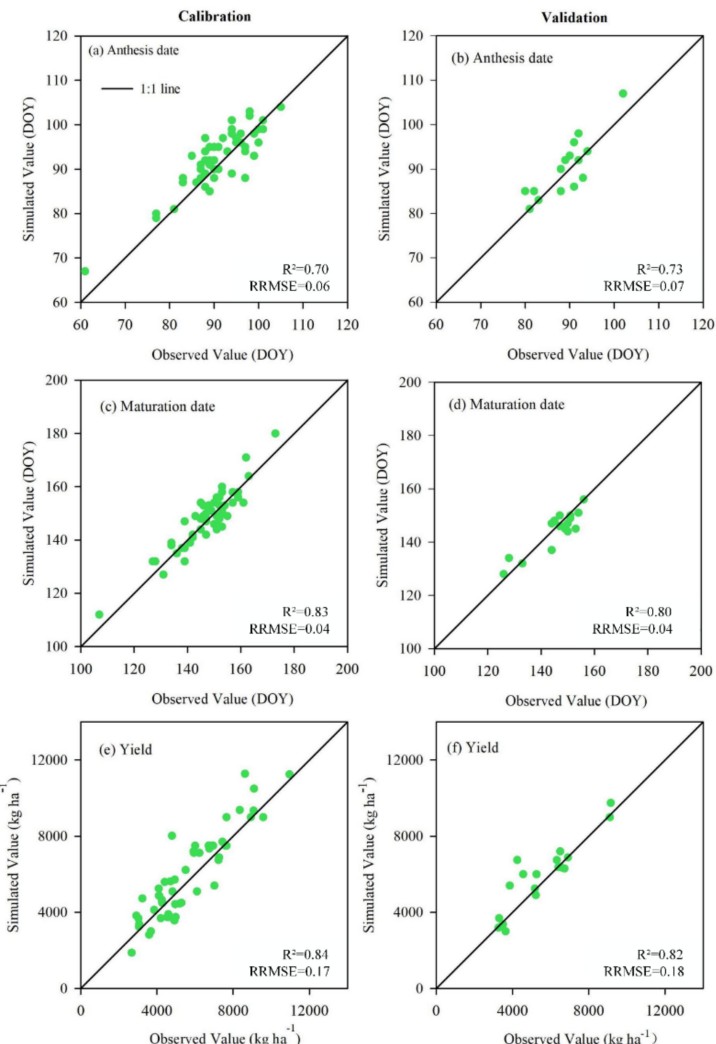

**Figure 7.** Comparison of the observed and simulated values of anthesis date, maturity date and yield for spring wheat for calibrated and validated processes.

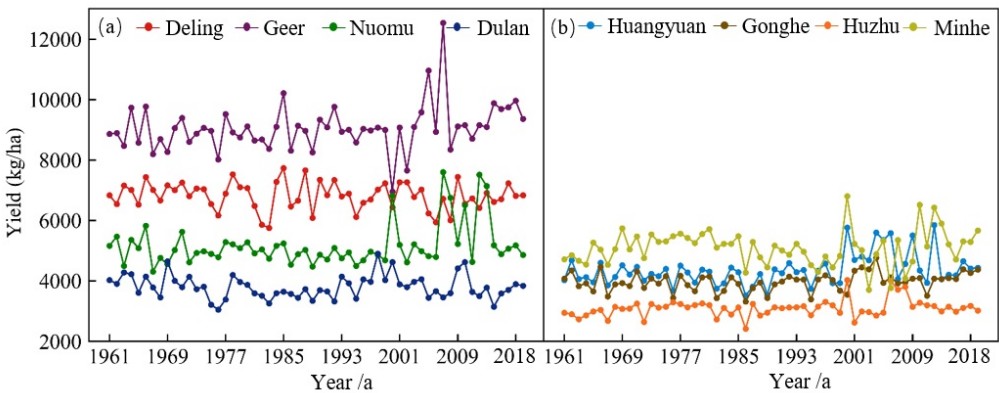

**Figure 8.** The annual variations of simulated (climatic) yields over 1961–2018 for the 8 selected sites. ((**a**) The 4 sites in the figure are located in the Qaidam Basin, and (**b**) the 4 sites in the figure are located in eastern Qinghai Province).

**Table 4.** The correlation coefficient values between spring wheat yield ($LAI_{max}$) vs. multi-scalar SPEI and $SMDI_{0-10}$, The numbers 1–6 indicate the timescale of drought index from 1 to 6 months, respectively. (** represents $p$-value $\leq 0.05$, and * represents $p$-value $\leq 0.1$).

| Growth Index | Index / Month \ Timescale | SPEI | | | | | | $SMDI_{0-10}$ | | | | | |
|---|---|---|---|---|---|---|---|---|---|---|---|---|---|
| | | 1 | 2 | 3 | 4 | 5 | 6 | 1 | 2 | 3 | 4 | 5 | 6 |
| LAImax | Mar. | −0.21 | −0.26 | −0.21 | −0.20 | −0.22 | −0.25 | 0.25 ** | 0.21 * | 0.17 | 0.12 | 0.04 | −0.01 |
| | Apr. | −0.14 | −0.21 | −0.25 | −0.22 | −0.22 | −0.24 | 0.32 ** | 0.32 ** | 0.27 ** | 0.23 * | 0.18 | 0.10 |
| | May | 0.00 | −0.07 | −0.13 | −0.16 | −0.15 | −0.15 | 0.35 ** | 0.36 ** | 0.35 ** | 0.32 ** | 0.28 ** | 0.22 * |
| | Jun. | −0.03 | 0.00 | −0.05 | −0.10 | −0.12 | −0.11 | 0.24 * | 0.31 ** | 0.32 ** | 0.32 ** | 0.29 ** | 0.26 ** |
| | Jul. | 0.11 | 0.05 | 0.04 | 0.01 | −0.02 | −0.04 | 0.09 | 0.21 * | 0.25 ** | 0.27 ** | 0.27 ** | 0.26 ** |
| | Aug. | −0.20 | −0.06 | −0.05 | −0.02 | −0.04 | −0.07 | 0.01 | 0.06 | 0.14 | 0.20 | 0.22 * | 0.23 * |
| Yield | Mar. | 0.07 | 0.05 | 0.11 | 0.16 | 0.12 | 0.06 | 0.22 * | 0.19 | 0.17 | 0.16 | 0.12 | 0.09 |
| | Apr. | 0.04 | 0.02 | 0.00 | 0.02 | 0.03 | 0.00 | 0.30 ** | 0.26 ** | 0.24 * | 0.22 * | 0.20 | 0.15 |
| | May | 0.16 | 0.13 | 0.11 | 0.10 | 0.12 | 0.12 | 0.37 ** | 0.36 ** | 0.33 ** | 0.32 ** | 0.30 ** | 0.26 ** |
| | Jun. | −0.11 | 0.04 | 0.03 | 0.03 | 0.03 | 0.05 | 0.30 ** | 0.32 ** | 0.32 ** | 0.30 ** | 0.29 ** | 0.27 ** |
| | Jul. | 0.17 | 0.05 | 0.09 | 0.10 | 0.09 | 0.09 | 0.14 | 0.25 ** | 0.27 ** | 0.28 ** | 0.27 ** | 0.26 ** |
| | Aug. | 0.02 | 0.12 | 0.04 | 0.10 | 0.11 | 0.11 | 0.07 | 0.11 | 0.20 | 0.22 * | 0.23 * | 0.23 * |

In order to better describe the correlations between drought indices and spring wheat growth-yield characteristics, a threshold value of $r = 0.21$ was selected, which represents a $p$-value $\leq 0.05$, below which it was assumed that there were no close connections between drought indices and crop growth. Since correlations between ($LAI_{max}$ or climatic yield) of spring wheat and $SMDI_{10-40}$ were mostly negative; therefore, the further analysis is meaningless.

The number of sites with $r > 0.21$ for SPEI and $SMDI_{0-10}$ at the 1- to 6-month timescales vs. spring wheat ($LAI_{max}$) climatic yield are presented in Table 5. It showed that (1) From SPEI, there were more sites with $r > 0.21$ from June to August (heading to maturity periods of spring wheat) (8/48) and the total number of sites with $r > 0.21$ between SPEI and spring wheat ($LAI_{max}$) climatic yield at the 1- to 6-month timescale during spring wheat growth period was minor (0 to 6 out of 48); (2) From SMDI, there were more sites with $r > 0.21$ from April to June (seedling emergence to heading period of spring wheat) ($LAI_{max}$, 6–11, climatic yield, 24–29). At the 2-month timescale, there was the largest correlation for $LAI_{max}$ (23/48) or climatic yield (13/48) vs. $SMDI_{0-10}$; (3) By comparison, the total site number of $r > 0.21$ correlations between ($LAI_{max}$) climatic yield and $SMDI_{0-10}$ were much larger than for SPEI, indicating better connections between spring wheat growth yield and $SMDI_{0-10}$. Therefore, $SMDI_{0-10}$ was a better index for denoting drought effects on spring wheat yields than SPEI and $SMDI_{10-40}$; and (4) Based on the site number between correlations of climatic yield and $SMDI_{0-10}$ with $r > 0.21$, there was the largest number of sites in May (11/48) and at the 2-month timescale (10/48); therefore, May was the key month and 2 months was the key timescale of SMDI for denoting the key parameters of preventing drought during spring wheat growth periods.

*3.4. Variations of Yield Reduction Rate*

From the above results, the 2-month $SMDI_{0-10}$ timescale in May is a key parameter to investigate the impacts of droughts on spring wheat yields; therefore, the data for specific parameters ($SMDI_{0-10}$, lagged by 2 months in May) are used for identifying normal and drought years, the average yield of normal years is used as the reference yield to further calculate the yield reduction rate in drought years.

The annual variations of the yield reduction rate and $SMDI_{0-10}$ in May for the eight sites are shown in Figure 9. In the severe drought years recorded in 1961, 1976, 1980, 1995, and 1999, all eight sites showed severe yield reduction rates (−4% to −31%). The variations of $SMDI_{0-10}$ were generally more consistent with the spring wheat yield reduction rate before 2001 for the $r$-values between production reduction rate and $SMDI_{0-10}$ in May over 1961–2000 were higher than that in 1961–2018 (0.21–0.40, except the Geer site). However, the consistent extent of yield reduction rate with $SMDI_{0-10}$ decreased after 2000 because Qinghai province has suffered extreme drought since its meteorological record, and al-

though after 2000, the drought was relieved and tended to be wetter, the yield did not increase accordingly.

**Table 5.** The number of stations with r > 0.21 between pairs of spring wheat growth indices (LAImax or climatic yields) and drought indices (SPEI and SMDI0–10 at the 1- to 6-month timescales) over 1961–2018. (The sum of the unconditional counts for each row or column is 48). Growth index.

| Index | | SPEI | | | | | | | $SMDI_{0-10}$ | | | | | | |
| Month \ Timescale | | 1 | 2 | 3 | 4 | 5 | 6 | Sum | 1 | 2 | 3 | 4 | 5 | 6 | Sum |
|---|---|---|---|---|---|---|---|---|---|---|---|---|---|---|---|
| $LAI_{max}$ | Mar. | 0 | 0 | 1 | 0 | 1 | 1 | 3 | 4 | 5 | 3 | 2 | 2 | 2 | 18 |
| | Apr. | 0 | 0 | 0 | 0 | 0 | 1 | 1 | 5 | 5 | 5 | 4 | 3 | 2 | 24 |
| | May | 0 | 0 | 0 | 0 | 1 | 0 | 1 | 4 | 5 | 5 | 5 | 5 | 5 | 29 |
| | Jun. | 0 | 0 | 0 | 0 | 0 | 0 | 0 | 4 | 4 | 5 | 5 | 5 | 5 | 28 |
| | Jul. | 0 | 0 | 1 | 0 | 1 | 1 | 3 | 1 | 2 | 2 | 4 | 4 | 5 | 18 |
| | Aug. | 1 | 0 | 0 | 1 | 0 | 1 | 3 | 0 | 2 | 1 | 2 | 3 | 3 | 11 |
| | Total | 1 | 0 | 3 | 3 | 4 | 5 | | 18 | 23 | 21 | 22 | 22 | 22 | |
| Yield | Mar. | 1 | 0 | 0 | 0 | 0 | 0 | 1 | 2 | 2 | 0 | 0 | 0 | 0 | 4 |
| | Apr. | 0 | 0 | 0 | 0 | 0 | 0 | 0 | 4 | 2 | 2 | 1 | 0 | 0 | 9 |
| | May | 0 | 0 | 0 | 0 | 0 | 0 | 0 | 2 | 4 | 1 | 2 | 1 | 1 | 11 |
| | Jun. | 1 | 1 | 1 | 1 | 1 | 1 | 6 | 1 | 1 | 1 | 1 | 1 | 1 | 6 |
| | Jul. | 0 | 1 | 1 | 1 | 1 | 1 | 5 | 0 | 1 | 1 | 1 | 1 | 1 | 5 |
| | Aug. | 1 | 0 | 1 | 1 | 1 | 1 | 5 | 0 | 0 | 0 | 1 | 1 | 1 | 3 |
| | Sum | 3 | 2 | 3 | 3 | 3 | 3 | | 9 | 10 | 5 | 6 | 4 | 4 | |

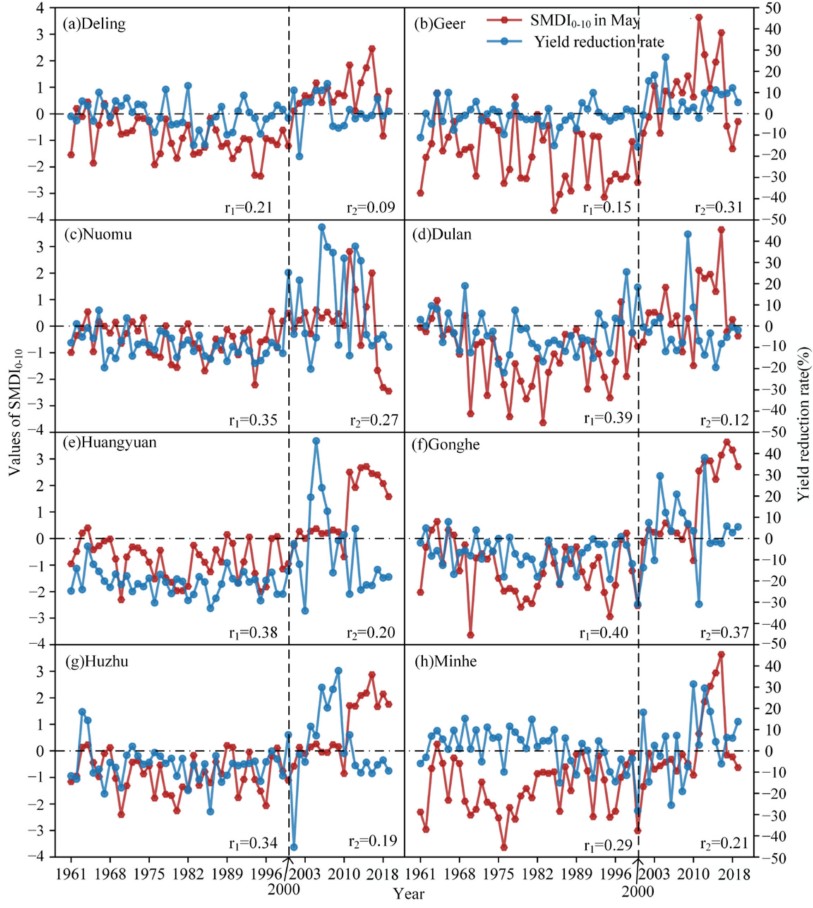

**Figure 9.** Annual Variations of yield reduction rate vs. 2-month-timescale $SMDI_{0-10}$ in May at the selected 8 sites over 1961–2018. $r_1$ and $r_2$ mean the Pearson correlation coefficients between production reduction rate and $SMDI_{0-10}$ in May in 1961–2000 and 1961–2018.

The relationships between linear slopes of 2-month $SMDI_{0-10}$ in May and the yield reduction rate at the eight selected sites are illustrated in Figure 10. The linear slopes of 2-month $SMDI_{0-10}$ in May and yield reduction rate ranged from $-0.04/a$ to $0.05/a$ and from $-0.001$ kg/a to $0.005$ kg/a, respectively. Additionally, the linear slopes of 2-month $SMDI_{0-10}$ in May in seven out of eight sites were positive, indicating that the climate conditions have a tendency to become wet from 1961 to 2018. Moreover, there was a considerable linear relationship between them with $r = 0.85$, suggesting that the trend of $SMDI_{0-10}$ in May was greatly consistent with the trend of spring wheat yield reduction rate from 1961–2018.

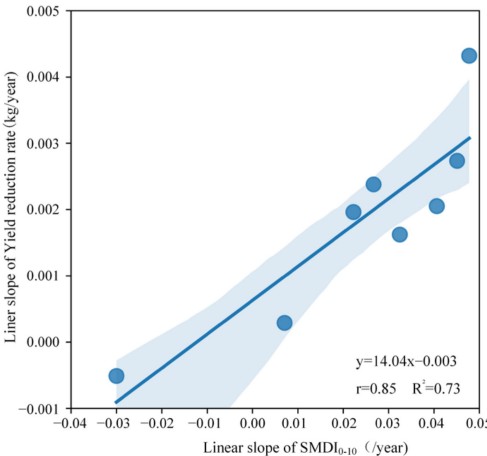

**Figure 10.** Variations in the linear slopes of spring wheat yield reduction rate vs. the 2-month $SMDI_{0-10}$ in May. (The shaded area represents 95 confidence interval).

## 4. Discussions

### 4.1. The Relationship between SPEI and SMDI

Meteorological drought is usually the first stage of a drought event, acting as starting point or driver of agricultural drought, and the different types of drought are interrelated, but with some spatial variability and temporal phase differences. Wang and Duan [59] pointed out that the degree of meteorological drought was more severe than agricultural in terms of degree, and agricultural drought lagged behind meteorological drought. Sims et al. [60] proved that there was a correlation between meteorological drought indices (PDSI and SPI) and soil moisture. SMDI takes into account more variables (such as evapotranspiration, soil properties, and root depth) than SPEI. Yared et al. [61] found that SPEI-3 showed a higher correlation with the agricultural drought index SMDI, and SMDI showed a delay compared to SPEI when comparing the drought start dates indicated by historical droughts. Fan et al. [62] pointed out that less precipitation in arid areas was almost the only source of surface soil moisture, and higher evapotranspiration would quickly evaporate it. The correlations between meteorological drought indices and soil moisture were only reflected in the shallow soil that can receive precipitation. This study similarly indicated that SPEI generally had higher correlations with $SMDI_{0-10}$, which lagged 1 or 2 months from May to August, but the correlations in March–April were not significant because the rainfall at the eight sites was concentrated in May–August but less in March–April, which may be not enough to supplement groundwater. The relationship between different drought indices has spatial–temporal variability and should be investigated for specific soil, climate and region conditions.

### 4.2. Better Drought Index for Monitoring Drought during Spring Wheat Growth

A number of drought indices could be applied to investigate drought conditions during the spring wheat growth period. Among them, soil moisture-related indices may be the more reliable index than the others. Previous research has shown that soil moisture (SM) variations (deficits or excesses) were the key factor affecting crop yield in rain-fed

agriculture [63,64]. Roots are vital organs for plants, and the effective use of resources from the soil was important for yield stability [65]. Our research showed that the correlations between spring wheat $LAI_{max}$/yields and $SMDI_{0-10}$ were better than with SPEI, indicating high spring wheat growth sensitivity to soil moisture deficiency to provide the necessary water and nutrients for crop growth, which will affect the morphogenesis, physiological processes and yield formation of the aboveground parts [66].

However, the performance of SMDI in revealing crops' drought responses varied with different depths. Zhang et al. [67] found that the root system of spring wheat in the dry land farming areas of central Gansu, China was the most at 0–10 cm in the seedling stage and the most at 10–30 cm in the flowering and maturity stages. Liao et al. [68] indicated that soil moisture could affect the growth of the root system and the planting area with lower soil moisture had smaller amounts of roots in the 0–20 cm soil layer. Sun et al. [69] found that the root system of spring wheat in the Southern Xinjiang was mainly distributed in the 0–40 cm soil layer and the root mass density and root length density decreased with depth. Jing et al. [70] suggested that 20 cm of irrigation could best overcome drought stress and that 40 cm of total water (precipitation and irrigation) was sufficient to maximize spring wheat production in the Canadian prairies. This research showed that $SMDI_{0-10}$ was closely related to the growth of spring wheat and should be paid more attention to, but the correlations between $LAI_{max}$/yield of spring wheat and $SMDI_{10-40}$ were not as good as $SMDI_{0-10}$. Although this research and others have found better performance of SMDI at 0–10 cm than SMDI at deeper depths and SPEI, how it performs for the other crops is unknown. Further studies are needed in furtherance to validate drought index performance in more conditions.

*4.3. The Key Month and Timescales Reflecting Crop Responses to Drought*

The impact of drought on spring wheat yields varied greatly due to the different drought levels in each growth period [71,72]. Kamali et al. [73] found a larger correlation between monthly SMDI and spring wheat yields from May to August. This period coincided with the reproductive stage of spring wheat development and was more susceptible to water stress. Wang et al. [74] pointed out that May to June were the heading and milking stages of spring wheat in the arid region and water availability had a very important influence on the growth of crops and the formation of grain yield. Yang et al. [75] indicated that wheat plant height growth was more sensitive in the jointing stage because the drought that occurred in the early stage of growth would force limited water and nutrients to the root system, promote root growth, and slow the growth of the shoots. Din et al. [76] pointed out that drought at the jointing stage significantly reduced the number of grains and decreased ear weight, which may be the main reason for the decline in yield. In this study, we found that the $SMDI_{0-10}$ in May (the jointing period of spring wheat) had the best correlation with the spring wheat yields, indicating that the soil moisture deficit at the jointing stage had a more adverse effect on spring wheat yield.

In addition, previous studies of drought on crops focused primarily on single-year events [77] but not on whether the multi-year drought affects crop growth in subsequent years. Peck and Adams [78] demonstrated the importance of analyzing individual years of drought in the context of previous years of drought. Continued soil drought would reduce wheat plant yield and marked changes in quality. Our research results showed that in normal years after successive droughts, yields would also be reduced, indicating that the drought was a slow accumulation process; therefore, a perennial drought would have certain impacts on crop yields in the subsequent years.

Yields may also be affected by other factors such as climate ones (frost, floods, etc.) and management ones (sowing, fertilization, and irrigation) but were not taken into account in this research. Studies on the multi-factor impacts on crop yield reduction are still needed in the future.

## 5. Conclusions

The temporal variations of SMDI and SPEI at different timescales (which reflect dry or wet conditions) during the growth period of spring wheat between 1961 and 2018 were investigated for the eight selected sites in Qinghai, China. SPEI varied more randomly than SMDI. The variations of SPEI were not so consistent as $SMDI_{0-10}$ and $SMDI_{10-40}$. SPEI had higher correlations with $SMDI_{0-10}$ at the lagged times of 1 and 2 months between May and August, with an increase in rainfall (0.35–0.68). $SMDI_{0-10}$ reflected actual drought events better (coincidence degree 43–82%) than SPEI (29–44%) and $SMDI_{10-40}$ (39–60%).

The DSSAT–CERES–Wheat model performed accurately in simulating the anthesis dates, maturation dates and meteorological yields of spring wheat. Through the correlation analysis of spring wheat yield vs. drought indices, $SMDI_{0-10}$ better reflected drought events during the spring wheat growth period ($0.21 < r < 0.37$), and 2-month (among six timescales) was the key timescale that best identified the relationship between spring wheat yields and $SMDI_{0-10}$. In addition, the drought in May had a severe impact on spring wheat growth and yields.

There were considerable correlations between the linear slopes of spring wheat yield reduction rate and linear slopes of $SMDI_{0-10}$ in May at the studied eight sites ($r = 0.85$). However, the correlation between production reduction rate and $SMDI_{0-10}$ was not as good as 1961–2000 ($0.20 < r < 0.40$) in 1961–2018 ($0.09 < r < 0.37$) due to the extreme drought in 2000. This study provides useful references for drought–resistance management.

**Supplementary Materials:** The following supporting information can be downloaded at: https://www.mdpi.com/article/10.3390/agronomy12071552/s1, Figure S1: Annual variations of monthly $P$, $ET_0$, $P − ET_0$ and $SD$ over 1961–2018 at 8 studied sites of Deling; Figure S2: Temporal variations of the monthly SPEI or SMDI at the 1- to 6-month timescales during the spring wheat growing seasonsat 8 studied sites of Deling; Table S1: Spring wheat growth period in different sites of Qinghai, China; Table S2: Genetic coefficients of spring wheat at the selected 8 sites; Table S3: Pearson correlation coefficients ($r$) between $LAI_{max}$ and drought indices (SPEI, $SMDI_{0-10}$ or $SMDI_{10-40}$) at selected 8 sites. (** represents $p$-value $\leq 0.05$, and * represents $p$-value $\leq 0.1$); Table S4: Pearson correlation coefficients ($r$) between the climatic yield and drought indices (SPEI, $SMDI_{0-10}$ or $SMDI_{10-40}$) at selected 8 sites. (** represents $p$-value $\leq 0.05$, and * represents $p$-value $\leq 0.1$).

**Author Contributions:** Methodology, M.H.; validation, N.Y.; formal analysis, N.Y.; investigation, A.B.; resources, Y.L.; data curation, M.H.; writing—original draft preparation, M.H.; writing—review and editing, Y.L., F.L. and I.H.; supervision, A.P. All authors have read and agreed to the published version of the manuscript.

**Funding:** This research was funded by [Key Research and Development Program of China] grant number [No. 2019YFA0606902]. And the APC was funded by [National Natural Science Foundation of China] grant number [No. 52079114].

**Institutional Review Board Statement:** No applicable.

**Informed Consent Statement:** No applicable.

**Conflicts of Interest:** The authors declare no conflict of interest.

## Abbreviations

SPEI: Standardized precipitation evapotranspiration index; SMDI, Soil moisture deficit index; DSSAT, Decision support system for agrotechnology transfer; GLDAS, Global land data assimilation system; SAT, Saturated moisture content; WP, Soil water content at the permanent wilting point; FC, Soil water content at the field capacity; SHC, Saturated hydraulic conductivity; $P$, Precipitation; $ET_0$, Reference crop evapotranspiration; GLUE, Generalized likelihood uncertainty estimation; $R_2$, Coefficient of determination; RRMSE, Relative root mean square error; $r$, Pearson correlation coefficient; $T_{max}$, maximum temperature; $T_{min}$, minimum temperature; $U_2$, wind speed measured at 2 m height; $RH$, Average relative humidity; n, sunshine hours.

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
