# Peer review of "Better Drought Index between SPEI and SMDI and the Key Parameters in Denoting Drought Impacts on Spring Wheat Yields in Qinghai, China"

_agronomy, doi:10.3390/agronomy12071552_

Round 1

Reviewer 1 Report

Title: OK

Abstract: OK

In abstract and keywords: SMDI is abbreviated as soil warer deficit index, it should be soil moisture deficit index

Fourth line of Introduction, after irrigation give comma in place of full stop

Page 2, last paragraph first sentence, yields should be yield

Page 3 second paragph the second sentence, give supporting reference of the statement

Study area and selected sites: the eveaporation 1400-2200 mm plz check the data

What is the basis of taking soil water content in dept 0-10, 10-40, 40-100 and 100-200 cm? plz mention

Table number and figure numbers are placed casually, plz correct them as per the text.

Page 10: P and ET changed periodically, can not be justified only through time series plot. Periodicity can be tested through wavelet analysi, plz do the analysis to test wheter periodicity is there, otherwise rewrite/delete the statement

Talbe 6 mentioned in text in page 12 doesnot exits

Figure 8; give location name in a, b, c, d, e, f g h

What do the author want to say through the linear line of figure 10? Plz explain properly

Author Response

We have carefully considered all comments from the reviewers and revised our manuscript accordingly. The manuscript has also been double-checked, and the typos and grammar errors we found have been corrected. In attachment, we summarize our responses to each comment from the reviewers.

Reviewer 2 Report

The text needs to be written more simply.

Some sentences are long and have lost their essence.

The order and numbering  of tables and figures must be uniform and logical in the text.

There are a lot of technical errors in the text and references.

Author Response

请参阅附件
